# Effects of Coal Gangue on the Hydrochemical Components under Different Types of Site Karst Water in Closed Mines

Bin-bin Jiang [1], Kai-ming Ji [1,2,*], Dong-jing Xu [1,2], Zhi-guo Cao [1], Shao-kun Wen [2], Kun Song [3] and Li Ma [4]

1    State Key Laboratory of Water Resource Protection and Utilization in Coal Mining, Beijing 100083, China
2    College of Earth Sciences &Engineering, Shandong University of Science and Technology,
     579 Qianwangang Road, Huangdao District, Qingdao 266590, China
3    Shandong Jining Yun-he Coal Mine Co., Ltd., Jining 272100, China
4    Shaanxi Provinical Key Laboratory of Geological Support for Coal Green Exploitation, Xi'an University of
     Science and Technology, Xi'an 710054, China
*    Correspondence: jkm153609@163.com

**Abstract:** In order to explore the potential effects of abandoned coal mines on the water quality of Ordovician limestone aquifers, water-rock interaction simulations were conducted. After the closure of the coal mine, the karst water in the goaf area and the waste gangue had a geochemical reaction, and the above-mentioned water-rock process was simulated by an indoor static immersion experiment to explore the differences in the effect of different types of karst water on the dissolution of gangue. The basic water quality parameters pH, EC (electrical conductance), and ORP (oxidation-reduction potential) showed different trends in karst hydro-immersion solution and ultra-pure hydro-immersion solution; pH and EC had greater fluctuations in two sets of ultrapure hydro-immersion solutions, while ORP fluctuated more widely in three groups of karst hydro-immersion solutions. In addition, gangue minerals dissolved more significantly in bodies of water where limestone was added. The results of chemical component clustering showed that TDS (total dissolved solids) and EC were homopolymerized in each immersion solution, and subsequent correlation analysis showed that TDS and EC clusters were more significantly affected by mineral properties in ultrapure water-immersion solutions, and more affected by dominant ions in karst water-immersion solutions.

**Keywords:** coal gangue; mine goaf; water quality parameters; cluster analysis; ions correlations

## 1. Introduction

The Carboniferous–Permian coalfields widely distributed in North China are important areas producing raw coal in China, and their coal resource reserves account for about 60% of China's total amount [1]. During the historical evolution of geology, the North China Plate as a whole was uplifted and suffered from long-term weathering and erosion at the end of the Middle Ordovician. It was not until the Late Carboniferous that it fell down again to accept sedimentation [2]. In other words, most of the Carboniferous–Permian coal-bearing strata in North China were deposited directly on the Ordovician limestone, and the Silurian of the early Paleozoic era and the Devonian of the late Paleozoic era were missing. The Ordovician limestone aquifer in significant thickness has developed karst fissures and good water abundance, which is an important hydrogeological feature in North China coalfields [3–5]. Coal gangue is one of the main solid wastes during the mining and washing process of coal, and its amount accounts for about 15% of the raw coal output [6]. As coal gangue accumulates on the ground and occupies too many valuable land resources, it is widely used in goaf filling in China [7–9]. After the working face mining ends, with the rebound of groundwater, the underlying Ordovician limestone karst water will gradually submerge the goaves and cause a series of geochemical reactions with minerals in coal gangue, especially the rapid oxidation of trace pyrite, dissolution and neutralization of carbonates, hydrolysis of feldspar and mica, etc., as shown in Figure 1 [10].

These reactions will inevitably bring about the growth of major components in water such as sulfate, chloride, calcium, magnesium, potassium, sodium, etc., thus making the total dissolved solids (TDS) in water show an increasing trend and leading to the deterioration of the quality of Ordovician limestone aquifer water [11]. Because the mine was no longer used after the mine is closed, the pollution of groundwater induced by the mine water in the goaf was often ignored. In fact, the detained mine water in the goaf may pollute the deep groundwater, which is far more hidden and serious than the pollution of the discharged mine water [12–14], which seriously affected the important water demand of urban industrial and mining enterprises. Due to the impact of human mining activities, the change in groundwater quality has always been a matter of worldwide concern [15].

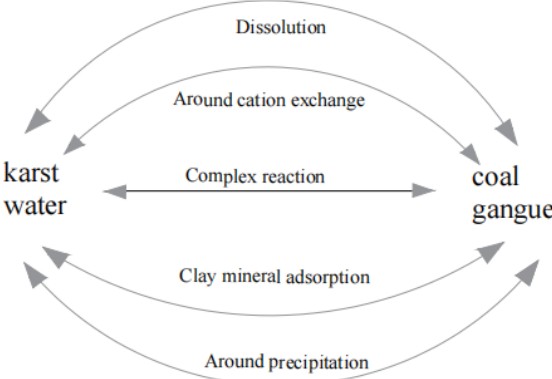

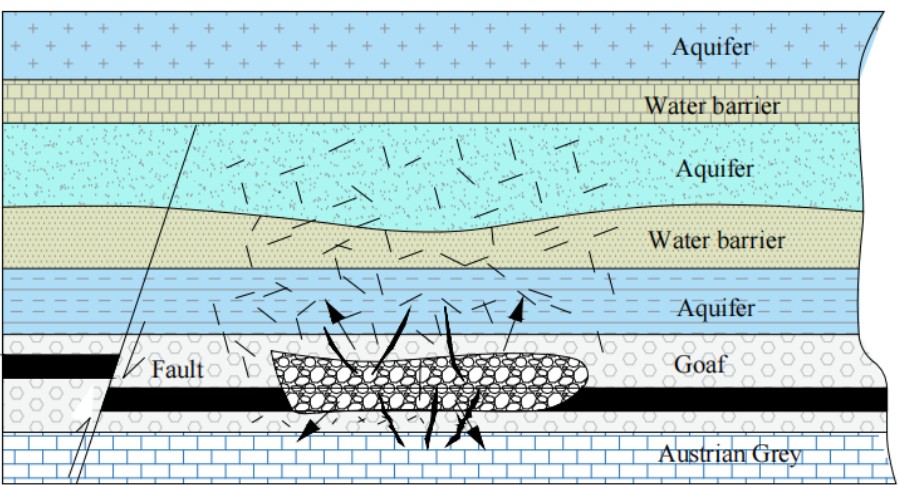

**Figure 1.** Schematic diagram of water-rock interactions in the mining area.

The static immersion test is a commonly used method to evaluate the release potential of the chemical components of solid materials. It can not only effectively solve the problem of water sample extraction in closed-pit mine goaves, but also facilitate the simulation of field conditions [16–19]. Moreover, the initial concentration and the release law of different chemical components in water over time are analyzed by changing the experiment parameters (such as pH, immersion temperature, solid–liquid ratio, immersion time, etc.). At present, many scholars have conducted much work on water pollution caused by wastewater discharged from the mining area [20,21], but there is less research on the water pollution caused by backing mining, especially the impact of coal gangue on the quality of Ordovician limestone karst water [22]. The main purpose of this research was to study the effect of coal gangue immersion on different types of karst water hydro-chemical composition using experimental methods of water rock immersion. By analyzing the change law of basic water quality indicators before and after soaking and the correlation

between key basic water quality indicators and other water chemical components, the influence of coal gangue mineral leaching on water quality was explored, and then the main factors affecting the change of Ordovician limestone karst water quality in closed mines were determined. This study is of great significance to reveal the influence of coal gangue on the change mechanism of water quality in the mining area, and also guarantee the safety of the Ordovician limestone aquifer water in the mine area and the reasonable utilization of coal gangue in the goaf.

## 2. Experimental Section

### 2.1. Research Area

In this paper, the study area involves three coal mines (seen Figure 2), namely the Hong-qi coal mine, the Bai-zhuang coal mine, and the Zhai-zhen coal mine. The Hong-qi coal mine is located northeast of the Ju-ye coalfield. The coal-bearing strata are the marine-terrigenous facies deposition of the Carboniferous–Permian in North China, and the coal-bearing strata basement is Ordovician limestone. The Ordovician limestone is widely distributed in the coal mine area,; the main lithology is blue-gray thick limestone or greenish gray marl limestone, and the lower part is dolomitic limestone.

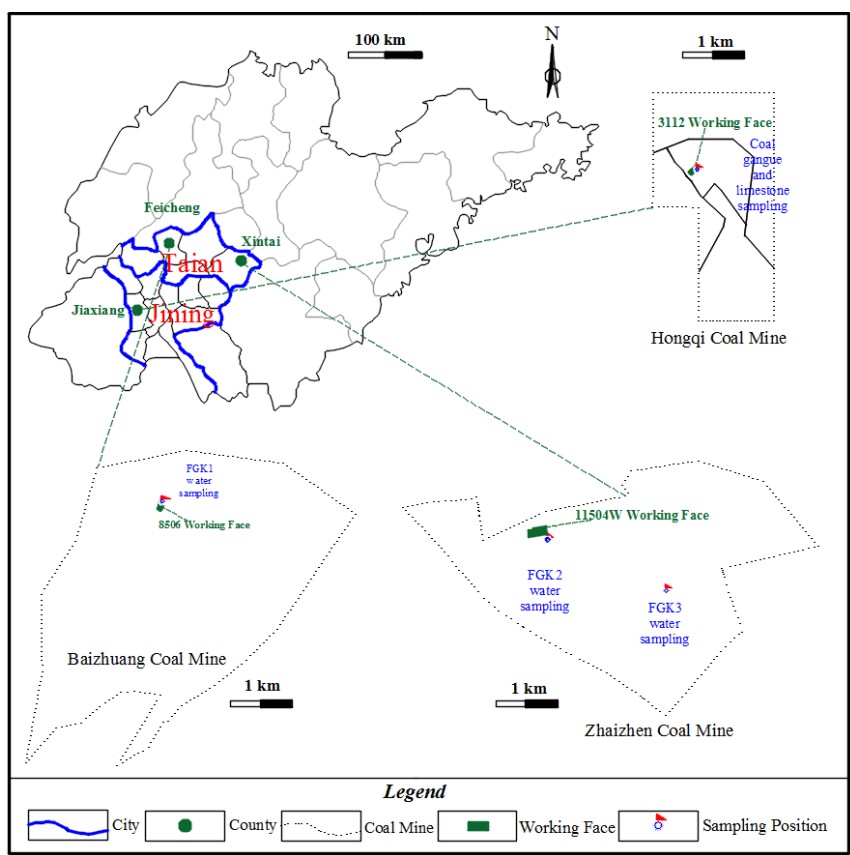

**Figure 2.** Geographic location of the research area in China.

Baizhuang Coal Mine is located in Feicheng, Shandong Province. The deposition of the whole area is stable, and the stratum thickness, lithology, and stratigraphic contact are basically the same as those in western Shandong. The main lithology of the Ordovician limestone aquifer in this area is limestone and dolomite. The Ordovician limestone aquifers are widely exposed on the surface, and directly receive the replenishment of atmospheric precipitation, with extremely high water yields and adequate recharge water [23]. The water chemical type is HCO3-Ca·Mg, the mineralization degree is 247~681 mg/L, belongs to low mineralization water, and the water quality belongs to class I–III according to GB/T 14848-2017 Groundwater Quality Criteria.

The Zhaizhen coal mine is located in Zhaizhen, Xintai City of Shandong Province; the upper part of the Ordovician limestone aquifer is thick-layered dense block limestone, the middle part contains vermiculite belts and nodules, and the lower part is brown or gray dolomitic thick-bedded crystalline limestone. The supply route is mainly affected by the small outcrop of atmospheric precipitation in the northern wing. The main form of Ordovician limestone aquifer water is high-pressure karst fissure static water [24]. The water chemical type is $SO_4$-K·Ca·Na·Mg. Its mineralization degree is 544~1252 mg/L, which is medium mineralization water. Water quality belongs to class II–IV according to GB/T 14848-2017 Groundwater Quality Standard.

### 2.2. Sampling and Preparation

Fresh coal gangue and Ordovician limestone samples from the Hongqi coal mine were collected as the main experimental solid samples. The gangue sample was collected from the 3112 working surface, and the weathered part and surface dust were cleaned up and placed in a polyethylene bag for later use. The mined gangue was crushed by a jaw crusher, and the block gangue of 0.45–3.2 mm was selected for water rock immersion experiment. Ordovician limestone (buried at a depth of 563 m) and screened gangue were washed and dried at 105 °C for 24 h. The above-mentioned treated limestone was then crushed and ground into a powder. Finally, the limestone powder and gangue samples were stored in an airtight container for experimental use.

Ordovician limestone aquifer water samples from the Baizhuang coal mine and Zhaizhen coal mine were collected as the main experimental immersion water samples. One Ordovician limestone aquifer water sample (roughly 3 L) was collected in the Baizhuang mine, and two Ordovician limestone aquifer water samples (roughly 3 L each) were collected from the Zhaizhen mine. All Ordovician limestone aquifer water samples were stored in polyethylene bottles and filtered in the laboratory for use. To simulate the ambient temperature of the goaf, a water sample mixed with gangue and mine water was left to stand in a 40 °C incubator.

### 2.3. Immersion Test

According to the analysis of the goaf of the Jia-iang Coal Mine by Wang Xiao-chen and Zhang Rong-ao [25,26], combined with the results of drilling experiments and the water storage coefficient of the rock formation, the relationship between the goaf of the coal mine and the mine water was restored in equal proportions, so that the mass ratio of water to rock was 10:1. The five impregnated solutions include one site in the Ordovician limestone aquifer water in Baizhuang, two Ordovician limestone aquifer water in Zhaizhen, ultrapure water, and a mixed solution of ultrapure water added by 10 g limestone powder (marked respectively FGK1, FGK2, FGK3, FGU and FGUL, see Table 1). Initial water quality parameters and hydro-chemical composition of different immersion solutions among the coal gangue immersion tests can be seen in Tables 1 and 2. The comparison test adopted a mixed solution of ultrapure water added with limestone powder, and the blank test uses ultrapure water instead. In order to realistically simulate the on-site temperature conditions of the goaf, the conical flask mixed with coal gangue and water was left to stand in a 40 °C constant temperature incubator. Three samples of each immersion solution were removed for parallel testing at the specified seven sampling times (1 d, 2 d, 4 d, 7 d, 7 d, 15 d, 25 d, and 35 d), and 15 test solutions of five types were collected for each experiment. Subsequently, the leaching solutions were filtered into polyethylene bottles using a 0.45 μm aperture microporous membrane and were stored at 4 °C until experimental analysis. Finally, coal gangue samples before soaking and after the last sampling time were ground into powder, and the chemical composition of the powder before and after soaking analysis using the ARL Perform X4200 X-ray Fluorescence (XRF) spectrometer was recorded (measurement method: wavelength dispersion sequence scanning type; analysis element range: O-U3; content range: ppm to 100%) [6,27].

**Table 1.** Design and Initial Water Quality Parameters of the Coal Gangue Immersion Tests under Different Immersion Solutions.

| Serial Number | Sample | Immersion Solutions | Buried Depth (m) | pH | ORP (mV) | TDS (mg/L) | EC (µS/cm) | DO (mg/L) |
|---|---|---|---|---|---|---|---|---|
| FGK1 | Fresh gangue | Baizhuang Ordovician limestone water | 250 | 7.64 | 176.6 | 480.3 | 819.0 | 7.23 |
| FGK2 | Fresh gangue | Zhaizhen Ordovician limestone water | 549.8 | 7.69 | 181.1 | 1251.7 | 1930.3 | 8.00 |
| FGK3 | Fresh gangue | Zhaizhen Ordovician limestone water | 395.07 | 9.57 | 22.1 | 543.7 | 932.3 | 9.11 |
| FGU | Fresh gangue | Ultrapure water | - | 5.83 | 277.2 | 2.9 | 5.48 | 9.22 |
| FGUL | Fresh gangue + limestone | Ultrapure water | - | 5.83 | 277.2 | 2.9 | 5.48 | 9.22 |

**Table 2.** Initial Hydro-chemical Composition of Different Immersion Solutions.

| Serial Number | $Na^+$ (µg/L) | $K^+$ (µg/L) | $Ca^{2+}$ (µg/L) | $Mg^{2+}$ (µg/L) | $Fe^{3+}$ (µg/L) | $Cl^-$ (mg/L) | $SO_4^{2-}$ (mg/L) | $NO_3$ (mg/L) |
|---|---|---|---|---|---|---|---|---|
| FGK1 | 50,800 | 1321 | 137,800 | 52,340 | 319.3 | 68.33 | 114.39 | 12.23 |
| FGK2 | 1,000,000 | 16,390 | 217,500 | 69,070 | 713.3 | 82.63 | 1224.58 | 3.46 |
| FGK3 | 1,000,000 | 11,770 | 29,210 | 27,660 | 92.41 | 6.92 | 368.97 | 1.44 |
| FGU | 0 | 0 | 0 | 0 | 0 | 0 | 0 | 0 |
| FGUL | 0 | 0 | 0 | 0 | 0 | 0 | 0 | 0 |

Potential Hydrogen (pH), Oxidation Reduction Potential (ORP), Conductivity (EC), Total dissolved solids (TDS), and Dissolved-oxygen (DO) were determined by a Hach Hydrolab multiparametric water quality analyzer. The concentrations of major cations (including potassium ions, sodium ions, calcium ions, magnesium ions, etc.) and anions (chloride ions, sulfate ions, nitrate ions, etc.) were determined, respectively, using the Agilent's 7500 inductively coupled plasma mass spectrometer and the ICS-600 ion chromatograph of the ThermoFisher company in the USA.

### 3. Results and Discussion

#### 3.1. Mineral Components Analysis of Coal Gangue

According to XRD analyses seen in Figure 3, the main minerals of coal gangue are kaolinite and quartz, and its secondary minerals are kaolinite, ankerite, smectite, sanidine, illite, and muscovite. This result is consistent with the analysis report of the Ju-ye coal exploration.

The contents of the primary and secondary constituents in the original coal gangue sample and the post-leached residues were determined by the XRF at the Shandong University of Science and Technology, which can be seen in Table 3. From Table 3 we can see that the major constituents in the original coal gangue sample were $SiO_2$ (33.36%), $Al_2O_3$ (22.87%), CaO (22.56%), MgO (9.58%), $Fe_2O_3$ (5.63%), $K_2O$ (2.72%) and $TiO_2$ (1.96%), and the minor constituents were $SO_3$, $Na_2O$, SrO, and MnO. As seen in Figure 4, the column chart shows the varying results of the major constituents from the original coal gangue sample to its post-leached residues. The XRF results showed that the main components in the residue after the leaching of coal gangue remained at lower values in the leaching solutions of FGK1 and FGU, compared with FGK2, FGK3, and FGUL, indicating that the relationship between the coal gangue matrix and the former aqueous solution of the reaction is relatively stronger than the latter solution.

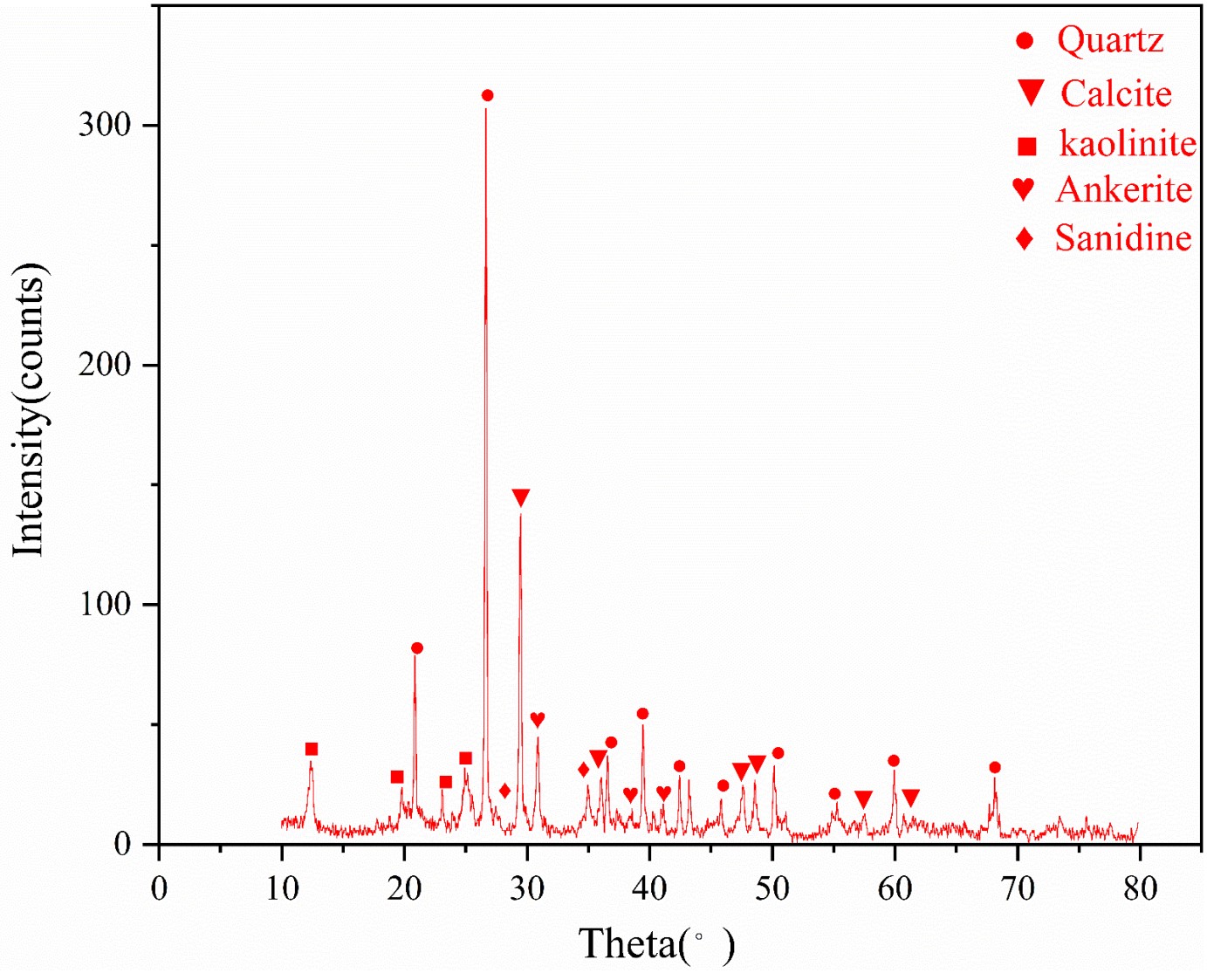

**Figure 3.** The Mineral XRD analysis results plot.

**Table 3.** Chemical Compositions of the Original Coal Gangue Sample and its Post-Leached Residues.

| Chemical Composition (%) | SiO$_2$ | Al$_2$O$_3$ | CaO | MgO | Fe$_2$O$_3$ | K$_2$O | TiO$_2$ | SO$_3$ | Na$_2$O | SrO | MnO |
|---|---|---|---|---|---|---|---|---|---|---|---|
| Initial sample (FO) | 33.359 | 22.872 | 22.557 | 9.576 | 5.632 | 2.719 | 1.964 | 0.655 | 0.454 | 0.127 | 0.084 |
| Post-leached residue of FGK1 | 38.596 | 23.806 | 17.188 | 8.324 | 5.503 | 3.011 | 2.045 | 1.184 | 0.205 | 0.07 | 0.069 |
| Post-leached residue of FGK2 | 31.895 | 20.246 | 27.121 | 9.772 | 5.293 | 2.392 | 1.668 | 1.053 | 0.295 | 0.122 | 0.081 |
| Post-leached residue of FGK3 | 32.421 | 21.727 | 25.102 | 10.171 | 5.084 | 2.552 | 1.708 | 0.785 | 0.256 | 0.113 | 0.082 |
| Post-leached residue of FGU | 31.313 | 19.671 | 28.363 | 9.973 | 5.246 | 2.333 | 1.686 | 0.966 | 0.207 | 0.154 | 0.087 |
| Post-leached residue of FGUL | 35.409 | 22.124 | 22.02 | 9.253 | 5.476 | 2.661 | 1.801 | 0.849 | 0.262 | 0.073 | 0.071 |

*3.2. Analysis of Basic Water Quality Parameters for Different Immersion Solutions Interacting with the Coal Gangue*

According to XRF results, water-rock interaction will change the mineral composition after immersion solution, and the chemical composition of immersion solution will also change accordingly. Thus, using the Hach Hydrolab Multi-Parameter Water Quality Analyzer, we obtained the pH, EC, ORP, TDS, and DO values for the coal gangue in different water solutions over time as displayed in Figure 5. The data for the immersion tests were reported as the averaged value of three parallel tests with maximum relative deviations of 6% (multiple water quality base parameters).

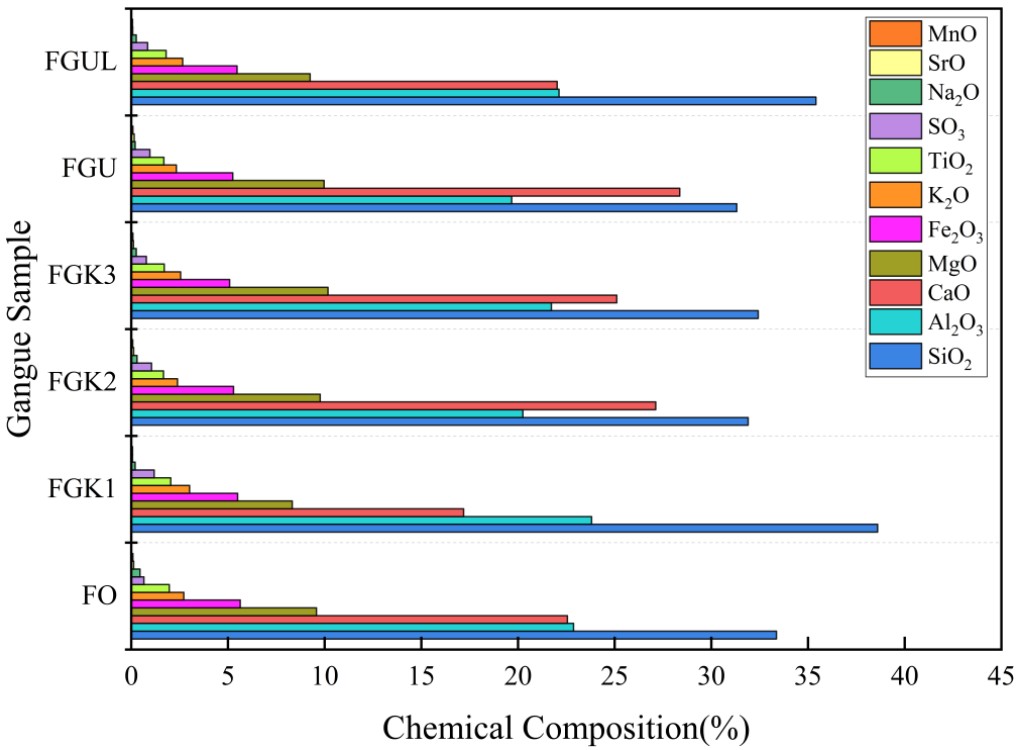

**Figure 4.** Column distribution diagram of the main components and their after-immersion residues in the raw gangue samples.

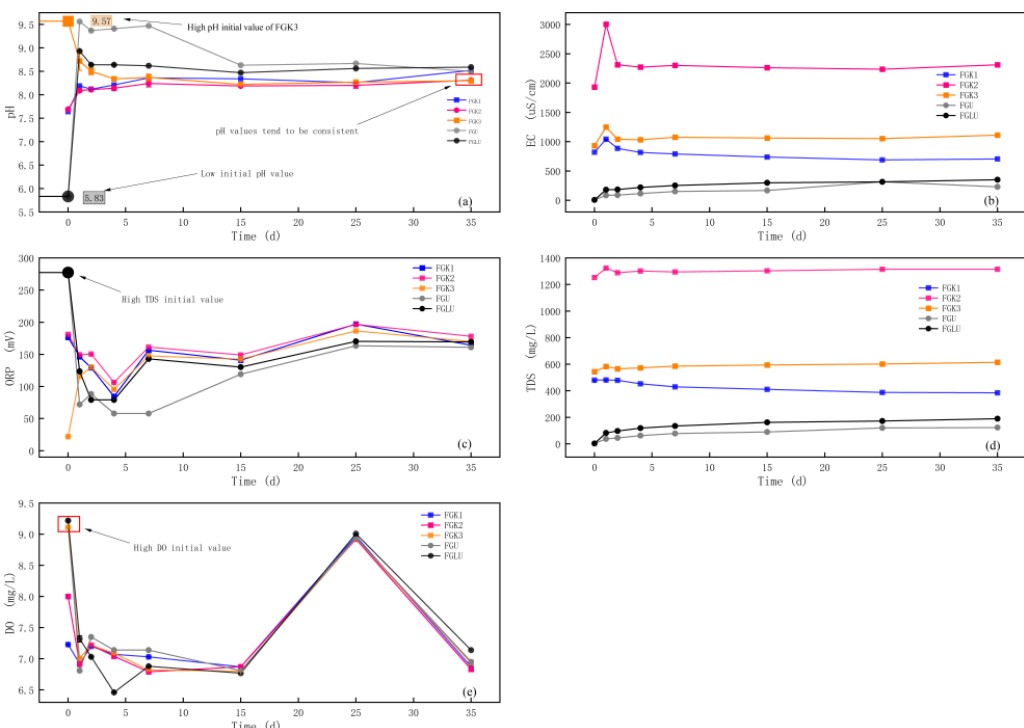

**Figure 5.** Basic water quality parameters of coal gangue in different immerse solutions over time. (**a.** Trend graph of pH changes of different immersion solutions over time; **b.** Trend graph of EC changes of different immersion solutions over time; **c.** Trend graph of ORP changes of different immersion solutions over time; **d.** Trend graph of TDS changes of different immersion solutions over time; **e.** Trend graph of DO changes of different immersion solutions over time).

Figure 5a presents that although the initial pH value are different, the pH values of all solutions of the five immersion solutions are concentrated between 8.1 and 8.7 after undergoing changes, maintaining a weakly alkaline condition. The pH change curves for FGK1 and FGK2 solutions showed roughly the same trends, rising from 7.64 to 8.35 and from 7.69 to 8.27 after a continuous rise within 7 d respectively, then finally remained steady to the end of the experiment. However, the pH change curve of the FGK3 solution decreased from 9.87 to 8.34 after a continuous decrease, and finally remained stable in a similar process to other solutions. This confirmed the different impact mechanisms of the site karst water, and the results also showed the buffering capability of coal gangue against acid and alkali [22,28]. There were significant differences in pH variations over time for FGU and FGLU, and although the pH change trends are roughly the same trends, they tended to have greater values for FGU solution than that of FGLU solution and took longer to recover the balance. In addition, the solutions both had much larger pH fluctuations than the other three site karst water solutions, despite having the lowest initial pH values. The different results between FGLU and FGU also showed the minerals of the limestone powder (mainly referring to carbonates) have a good buffer effect on the dissolution of coal gangue in water, controlling the interaction between the minerals of the fresh gangue and water to a great extent.

From Figure 5c we can see that the ORP change curves for all solutions showed roughly the same trends, although the initial ORP values varied greatly. It also shows that all ORP values for the five kinds of solutions changed greatly in their earlier stages by 4 d primarily, and then kept an undulant increase until the end of the experiment. The ORP values of all solutions were observed in the following order: FGK2 > FGK3 > FGK1 > FGLU > FGU, which was similar to the order for EC. In addition, although having the strongest initial ORP values, the two lab-configured water solutions (FGU and FGLU) both showed a gradually reduced degree of redox reactions of all solutions over time.

DO always depends on the temperature and the organic content in water [29]. Figure 5e showed that the DO values of all solutions had similar overall trends, indicating that the main interaction time between the minerals of coal gangue and water was roughly the same. The DO change curve of all solutions reached dissolved peaks with the same schedule, which indicated that DO change did not matter much with the type of water. However, the DO values of the FGLU solution were obviously lower than the other four solutions after 4 d, showing an obviously different effect of limestone powder on the water DO value.

The pH value of the five groups of immersion solutions remained between 8.1–8.7 after a long period of water-rock interaction, showing weak alkalinity, and the change of pH value demonstrated the buffering ability of gangue minerals to the changes of acid and alkali of the immersion solution. ORP changes occurred mainly within the first 4 d, with the consistent ordering of ORP and EC in the five groups of immersion solutions, but the ORP value after immersion in ultrapure water solutions was lower than the initial ORP value. DO in all solutions had a similar trend of change, reflecting that DO variations are independent of the chemical type of water.

### 3.3. Cluster Analysis of the Water Quality Parameters of Each Solution

Cluster analysis is a statistical analysis technique that divides the study subjects into relatively homogeneous groups. The analysis of water quality parameters showed that karst water and laboratory water use have different trends and are influenced by different factors. Therefore, we used the SPSS19.0 software for conventional water chemical components and water quality parameters of 14 indicators r cluster analysis, and used the R-type cluster tree diagram of each water body (Figure 5) to explore the correlation between each water quality parameter. In the Figure, the abscissa is the group distance and the ordinate is the variable. When the group distance is used to measure the relationship between variables in r-type clustering analysis, the smaller the group distance is, the closer the relationship is. According to the result of cluster analysis (Figure 5a,e) it can be seen that the water

body clustering effect is significantly different. Compared with the lab-configured water system, the distribution of the components in the karst water varies greatly, the correlation is overall poor, and there exist obvious differences, so the water body in clustering analysis used different groups of distance as the standard selection object.

It can be seen from Figure 6a, with a group distance of five as the selection criteria, TDS, $SO_4^{2-}$, EC and $F^-$ cluster together in the FGU water body, and the group distance between them is minimum and equal, indicating that these components are closely related and have the same influence. Mineral dissolution in FGU is mainly gypsum and mirabilite, so $SO_4^{2-}$ affects the changes of EC and TDS. The change of $F^-$ ion concentration is related to the dissolution of muscovite and fluorite, and may also be affected by the dissolution of mirabilite and gypsum [30].

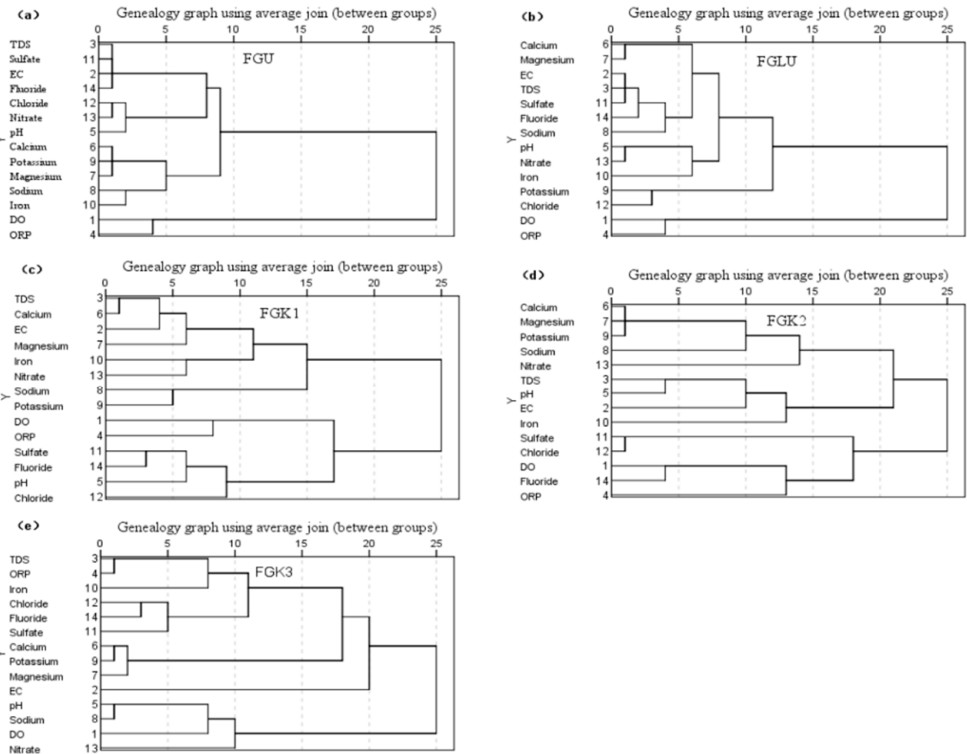

**Figure 6.** Correlation diagram of water quality indicators.

It can be seen from Figure 6b, with a group distance of five as the selection criteria, EC, TDS, $SO_4^{2-}$, $F^-$ and $Na^+$ in FGLU water body are classified into one category. The group distance between EC, TDS, and $SO_4^{2-}$ is minimum and equal, indicating that $SO_4^{2-}$ has the same influence on EC and TDS. Sodium and fluoride ions converge into one class, which indicates that the release of $Na^+$ ions affects the change of $F^-$ content, which may be because the two ions' sources overlap or are affected by other hydro-chemical actions, thus affecting the dissolution of gypsum with the changes of TDF and EC.

Figure 6c shows that TDS, $Ca^{2+}$, EC, and $Mg^{2+}$ in FGK1 water are grouped into groups with a group distance of 10 as the selection criteria. The group distance between TDS and $Ca^{2+}$ is the smallest, indicating that the relationship between these two components is the closest, and TDS changes are significantly affected by the initial high $Ca^{2+}$ concentration in FGK1 water. The clustering results indicate that the dissolution or cation exchange of clay minerals such as kaolinite affects the change of EC and then TDS.

As can be seen from Figure 6d, TDS, pH, EC, and Fe in the FGK2 water body are grouped together using a group distance of 10 as the selection criteria. The group distance between TDS and pH is the smallest, indicating that these two components are most closely related. The FGK2 clustering result indicates that the dissolution of pyrite in FGK2 water has a great influence on the changes in TDS, EC, and pH. According to studies by

Fuoco et al. and Fan et al. [31,32], the dissolution of pyrite affects the content of $SO_4^{2-}$, $H^+$ and $Fe^{3+}$ plasma in the water, which has the greatest influence on the change of EC, followed by the change of pH, and then the change of TDS.

As can be seen from Figure 6e that TDS, ORP, and Fe in the FGK3 water body are classified together using a group distance of 10 as the selection criteria. The group distance between TDS and ORP is the smallest, indicating a close relationship between these two components. The two fractions and the iron were clustered, which indicates that the dissolution of pyrite in FGK3 water has a great influence on the changes of TDS and ORP. According to studies by Fuoco et al. and Fan et al. [31,32], the dissolution of pyrite changes the ion content of $Fe^{3+}$ and $SO_4^{2-}$ in the water, affects the changes of ORP and TDS, and the influence degree of these two components is consistent. The EC is separately classified as a category in the FGU water bodies, indicating that this component is not closely related to other components and is less affected by other components.

The clustering of water quality parameters showed that TDS and EC were significantly correlated in all immersion solutions, meantime sulfate shows a large influence on both TDS and EC clustering. Furthermore, the analysis results also showed that there were differences in the factors affecting TDS and EC clustering. Therefore, the correlation analysis of TDS and EC was conducted to explore the influence of different hydro-chemical components on them.

*3.4. Main Influencing Factors Controlling the TDS and EC*

3.4.1. TDS Correlation Analysis of Each Immersion Solution

According to the hydro-chemical components change (as shown in Figure 7) of FGU solution in the blank control group, all ion content increased with TDS, but $Na^+$ and $SO_4^{2-}$ increases were relatively large and had high concentration in the water. It can be seen that $Na^+$ and $SO_4^{2-}$ ions in FGU immersion solution are the main hydro-chemical components that determine the salinity of the water body. According to the hydro-chemical components analysis results of the blank control group and the ion changes of different types of karst water, it was found that the ions that have a great influence on TDS are obviously different in each water body. In FGK1 and FGK2, $Na^+$, $Ca^{2+}$, $Mg^{2+}$, $SO_4^{2-}$, and $Cl^-$ have a great influence on the total dissolved solids; however, in an FGK3 water body, $Na^+$, $SO_4^{2-}$ and $Cl^-$ have a great influence on total dissolved solids. It can be seen that $Na^+$ and $SO_4^{2-}$ are the main hydro-chemical components determining the total dissolved solids in the water-rock interaction system. From these results, the influence of $Na^+$ and $SO_4^{2-}$ on the TDS should be considered in the analysis of the water's total dissolved solids. IBMSPSS statistical software was used to obtain Pearson correlation analysis between TDS in all aqueous solutions and major hydro-chemical ions (sodium, potassium, calcium, magnesium, iron, sulfate, chloride, nitrate, and fluoride), as shown in Figure 7e.

It can be seen from Figure 8a that TDS and $SO_4^{2-}$ have a strong correlation ($R^2 = 0.976$) in FGU water, indicating that the concentration and spatial distribution of $SO_4^{2-}$ play a key role in TDS. TDS with $Na^+$, $SO_4^{2-}$ and $Na^+$ were strongly correlated, indicating that dissolution of Mirabilite ($Na_2SO_4 \cdot 10H_2O$) had a high contribution to TDS change. The correlation between TDS and $Ca^{2+}$ and between $SO_4^{2-}$ and $Ca^{2+}$ was moderate, indicating that gypsum dissolution contributed to the increase of TDS. The correlation between TDS and $Cl^-$, $Na^+$ and $Cl^-$ was moderate, indicating that salt dissolution also contributed to the change of TDS.

It can be seen from Figure 8b that TDS and $SO_4^{2-}$ also have a similarly strong correlation ($R^2 = 0.967$) in FGU water, indicating that the concentration and spatial distribution of $SO_4^{2-}$ in FGLU water also play a key role in TDS. The correlation between TDS and $Na^+$, $SO_4^{2-}$ and $Na^+$ is extremely strong, indicating that dissolution of mirabilite in FGLU water contributes greatly to the change of TDS than that in FGU. This may be due to the dissolution of limestone and other calcium-bearing minerals, which increases the concentration of calcium ions, enhances the adsorption of cation alternation, and makes the dissolution of mirabilite easier. The TDS and $Ca^{2+}$, $Ca^{2+}$ and $SO_4^{2-}$ correlations achieve a strong degree,

in which the correlation increased in FGLU compared to FGU. It is shown that the dissolution of gypsum contributes more to the change of TDS, the dissolution of gypsum can be caused by the limestone dissolution making the calcium ions concentration in the water increase, according to the common ion effect, promoting the precipitation of gypsum, and the total dissolution amount of gypsum in the water body will also be increased accordingly, so it has a greater impact on the TDS change. The correlation between TDS and $Cl^-$, $Na^+$ and $Cl^-$ is weak, compared with FGU, and rock salt contributes less to the increase of TDS. It is possible that the salt dissolves more rapidly and the sodium concentration increases rapidly, and according to the common ion effect, rock salt dissolution is inhibited, so the contribution of rock salt to TDS decreases compared with FGU.

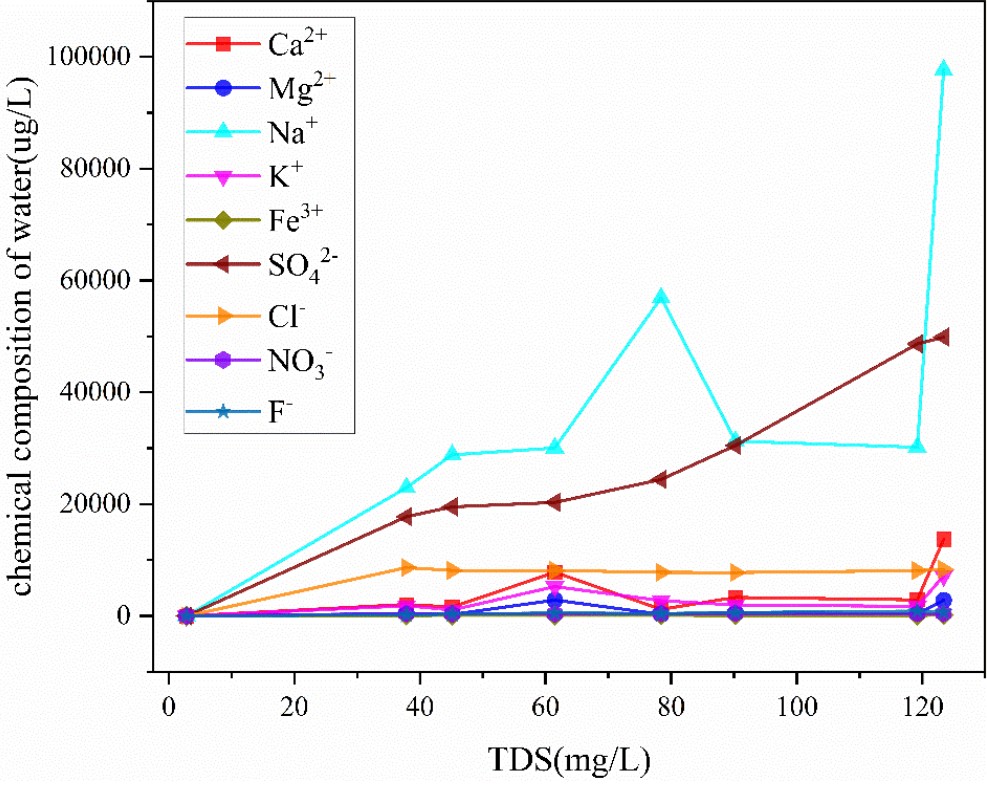

**Figure 7.** Plot of the hydro-chemical components of the FGU immersion solution change with TDS.

Figure 8c shows that the correlation between TDS and $SO_4^{2-}$ in FGK1 water body is −0.843, indicating that the dissolution of sulfate minerals has a negative effect on the increase of TDS and has a great influence. TDS is negatively correlated with sodium ions, and sodium ions are positively correlated with sulfate ions, indicating that mirabilite dissolution exists, but mirabilite dissolution is weak and has a negative effect on TDS increase. The strong correlation between TDS and $Ca^{2+}$ indicates that calcium concentration plays a key role in the change of TDS. The reasons for the change were analyzed as follows: the initial calcium ions concentration of the water sample was large, and calcium ions were absorbed by clay minerals in the initial stage of soaking. In addition, according to the common ion effect, the dissolution amount of gypsum in FGK1 water was small or insoluble, and the calcium ions concentration in the water decreased, and TDS decreased accordingly.

Figure 5 shows that TDS in FGK2 water gradually increases with the extension of soaking time. According to Figure 8d, TDS is negatively correlated with sodium and calcium ions in FGK2 water, and TDS is positively correlated with $SO_4^{2-}$, indicating that both sodium and calcium ions have a negative effect on the increase of TDS, while $SO_4^{2-}$ has a positive effect on the increase of TDS. The correlation between TDS and calcium ions is −0.053 greater than the correlation between TDS and sodium ions −0.383, indicating that sodium ions have a greater influence on TDS changes. The reasons for the change were

analyzed as follows: the original concentration of sodium and calcium ions in water was high, and sodium and calcium ions were absorbed by clay minerals in the early soaking stage. The decrease in ions concentration had a negative effect on the increase of TDS, which was consistent with the decrease of TDS in the early soaking stage. The concentration of sodium ions in the initial water is nearly five times that of calcium ions, sodium ions are preferentially adsorbed by clay minerals and larger adsorption amounts, so there is a greater negative correlation between sodium ions and TDS. In addition, according to the common ion effect, the higher sodium and calcium ions lead to mirabilite and gypsum being slightly or insoluble at the initial stage of soaking. In the middle and late soaking period, mirabilite and gypsum dissolved with the decrease of sodium ions concentration, and the increase of $SO_4^{2-}$ concentration had a positive effect on the increase of TDS, which was consistent with the increase of TDS in the middle and late soaking period.

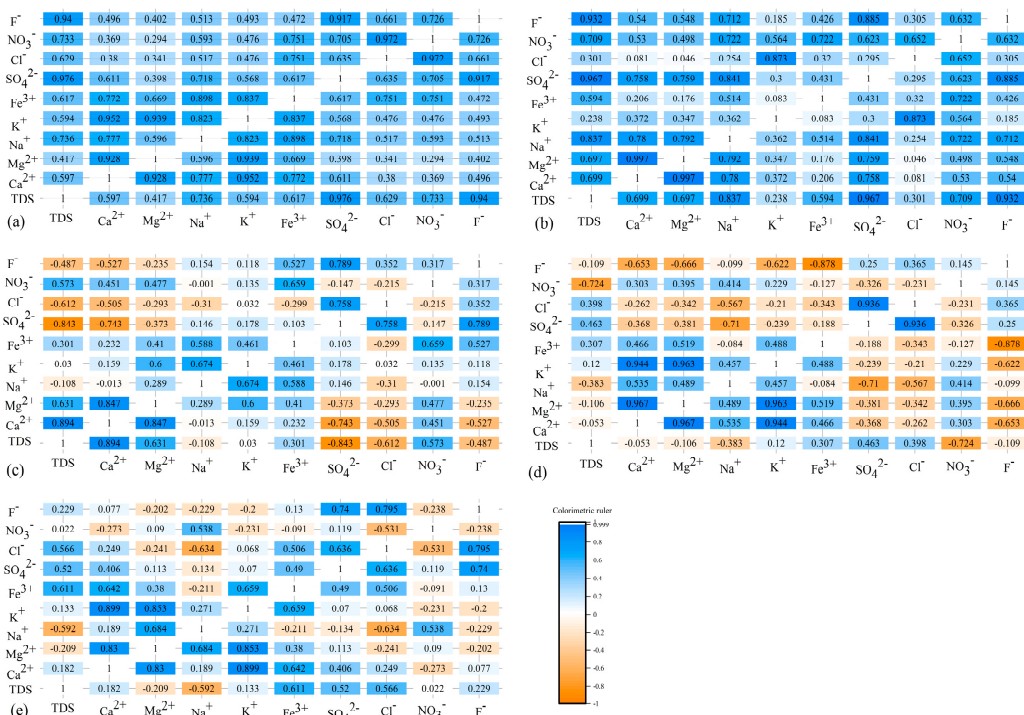

**Figure 8.** TDS and ions correlation plots. (**a**) TDS and ion-dependent heat map of FGU immersion solution; (**b**) TDS and ion-dependent heat map of FGLU immersion solution; (**c**) TDS and ion-dependent heat map of FGK1 immersion solution; (**d**) TDS and ion-dependent heat map of FGK2 immersion solution; (**e**) TDS and ion-dependent heat map of FGK3 immersion solution.

It can be seen from Figure 8e that in an FGK3 water body, TDS is negatively correlated with sodium ions ($R^2 = -0.592$), and TDS is positively correlated with sulfate ions ($R^2 = 0.52$) and reaches a moderate correlation degree. The reasons for the change were analyzed as follows: the initial sodium ions concentration in water was high. According to the common ion effect, the dissolution of mirabilite was inhibited, and the sodium ions were absorbed by clay minerals in the early soaking stage. Therefore, the decrease of sodium ions concentration in water had a negative correlation with TDS. Compared with FGK2, the negative correlation between sodium ions and TDS is larger, which may be due to the stronger alkalinity of FGK3 water and the stronger cation alternating adsorption [33], the larger adsorption amount of sodium ions, and the greater negative correlation. TDS is positively correlated with calcium and sulfate ions, and calcium ions and sulfate ions also have a positive correlation, indicating that gypsum dissolution had a positive effect on the increase of TDS, which was consistent with the increase of TDS in the early stage of soaking.

The correlation analysis heat maps of water chemistry component showed a significant positive relationship between TDS elevated and sulfate ions. In the ultrapure water immersion solution, the correlation between TDS and mineral dissolution was more significant, while the initial sodium and calcium ions in the karst water immersion solution obviously hindered the dissolution of sulfate minerals. The higher the initial ion concentration, the greater the interference with the dissolution of sulfate minerals.

### 3.4.2. EC Correlation Analysis of Each Immersion Solution

The determination of conductivity reflects the amount of ions content in water, and its size depends on the composition and content of ions in aqueous solution, as well as the temperature and viscosity of aqueous solution. Pearson correlation analyses between FGU and FGLU solutions EC and major water chemical ions (sodium, potassium, calcium, magnesium, iron, sulfate, chloride, nitrate, and fluoride) are shown in Figure 9.

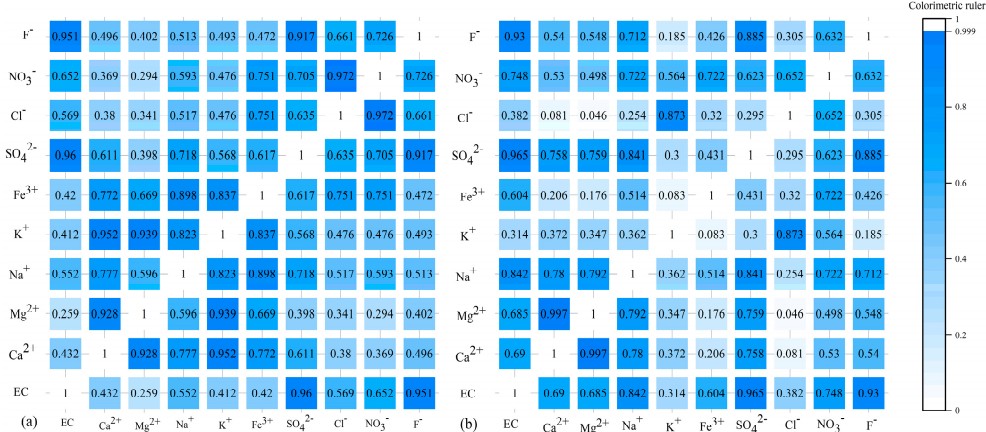

**Figure 9.** EC and ions correlation plots of FGK1 and FGLU immersion solution. (**a**) EC and ion correlation heat map of FGU immersion solution; (**b**) EC and ion correlation heat map of FGLU immersion solution.

A comparison of the two plots revealed that the EC and $SO_4^{2-}$ correlation values were close in FGU and FGLU aqueous solutions, indicating that $SO_4^{2-}$ contributed the same EC to both groups of solutions. EC and $Na^+$ had a moderate correlation of 0.552 in FGU and a very strong correlation of 0.842 in FGLU, and apparently a higher correlation between EC and $Na^+$ in FGLU water, which may be related to the fact that the addition of quicklime in FGLU promoted the dissolution of nitrate. In addition, the correlation between FGLU water EC and calcium, magnesium, and iron ions reached a strong correlation, greater than that between FGU water EC and calcium, magnesium, and iron ions, which was related to the addition of quicklime in FGLU water, which improved the concentration of calcium ions in water and promoted the dissolution of gypsum, calcite, pyrite, and other minerals.

According to the analysis of the blank control group, $Na^+$ and $SO_4^{2-}$ play a leading role in EC change in the pure water immersion system, but there are obvious differences in EC change in the karst water system affected by karst water type. In Figure 10a, EC and calcium ions correlation is higher in FGK1 water bodies compared with EC and sodium ions, and calcium ions in the initial solution reach 2.7 times the concentration of sodium ions, which has a greater impact on EGK1 solution EC. However, EC was negatively correlated with $SO_4^{2-}$, which was weakly correlated with the dissolution of sulfate minerals in the FGK1 solution. Figure 10b shows that the correlation between EC and cation in an FGK2 water body is not high, which is consistent with the TDS analysis, because mirabilite and gypsum are weakly dissolved in FGK2. However, the initial concentration of sodium ions in FGK2 is much higher than that of calcium ions, so sodium ions had a greater influence on EC than calcium ions. Figure 10c shows that EC has a negative phase relationship with $Na^+$ and $SO_4^{2-}$ in the FGK3 water body. According to the common ion effect, a higher

initial sodium ions concentration affects the dissolution of mirabilite and gypsum, while clay mineral adsorption leads to a decrease in sodium ions concentration, which has a negative influence on EC.

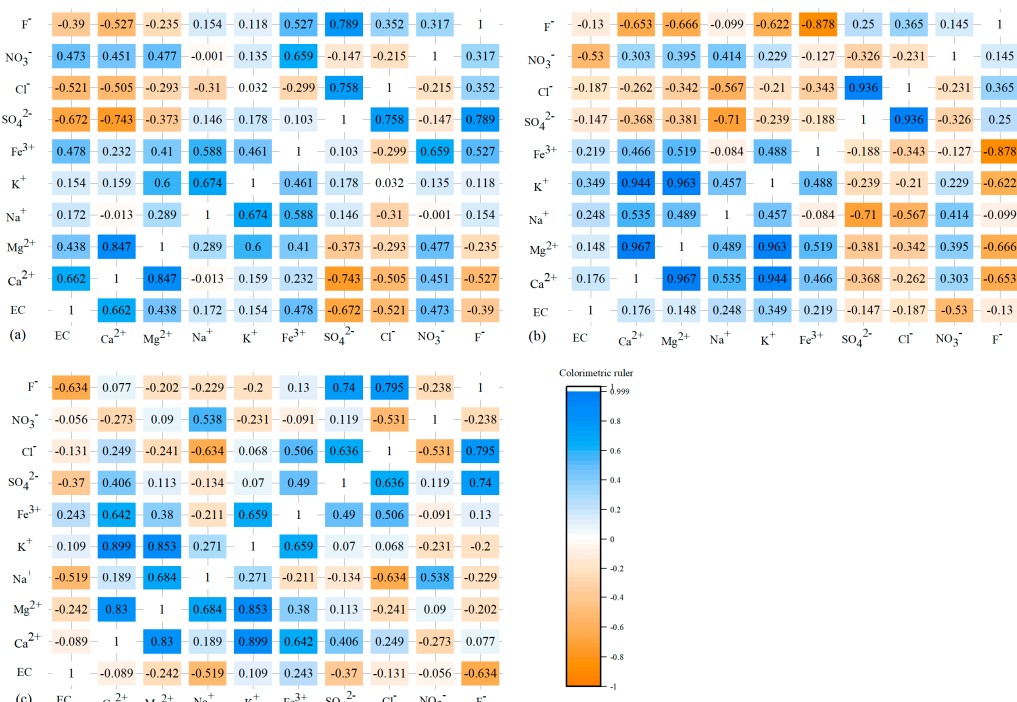

**Figure 10.** EC and ions correlations plots of FGK1, FGK2, and FGK3 immersion solutions.

Similar to TDS, sulfate has a significant effect on EC changes. Sulfate is more influential in the ultrapure water-immersed solution of limestone, and also proves that limestone promotes the dissolution of gangue minerals. In karst water-immersed solutions, $Na^+$ and $Ca^{2+}$ hinder the rise of EC due to the common ion effect, and $Ca^{2+}$ has a greater influence in FGK1, while $Na^+$ has a more significant effect in FGK2 and FGK3.

## 4. Conclusions

(1) After gangue is dissolved, no matter what the initial water chemical type, the pH always tends to develop in the direction of acid–base balance. In contrast to ultrapure water-immersion solutions, the initial ions in karst water always hinder the dissolution of gangue minerals, especially sulfate minerals. However, the addition of limestone to the ultrapure water-immersion solution can promote the dissolution of gangue minerals.

(2) TDS and EC were significantly correlated in all immersion solutions. Sulfate is a key influencing factor in TDS and EC clustering. Elevated TDS and EC are positively related to sulfate minerals. In the ultrapure water immersion solution, the correlation between TDS and mineral dissolution was more significant, while the initial sodium and calcium ions in the karst water immersion solution obviously hindered the dissolution of sulfate minerals. The higher the initial ion concentration, the greater the interference with the dissolution of sulfate minerals.

(3) Studies have found that karst water types affect the dissolution of gangue minerals. However, due to the composition and minerals of coal gangue being very complex, its decisive hydro-chemical role needs to be further determined, and the contribution of each class hydro-chemical component in the process should be specifically analyzed in the follow-up research.

**Author Contributions:** All authors contributed to the study conception and design. B.-b.J. was responsible for leading the planning, execution, and review of research activities. K.-m.J. was the main author of the manuscript and has made major contributions to data collection, data analysis, content analysis. D.-j.X. was mainly responsible for guiding the experimental protocol. He was also the main funder of the study. Z.-g.C. was mainly responsible for the detection of samples and the proofreading of experimental data. S.-k.W. has contributed to the operation in experiment, especially the geological work and operation in experiment. K.S. was responsible for the production and treatment of the field water and rock samples for this experiment. L.M. is mainly responsible for visualization, literature collection, and drawing the charts. All authors have read and agreed to the published version of the manuscript.

**Funding:** This work was supported by the projects of the Natural Science Foundation of Shandong Province, China (ZR2022MD101), Open Fund of State Key Laboratory of Water Resource Protection and Utilization in Coal Mining (Grant No. WPUKFJJ201909), Science and Technology Research Directive Plans of China National Coal Association (Grant MTKJ2018-263), and Fund project of Shaanxi provinical key laboratory of geological support for coal green exploitation (Grant DZBZ2020-01).

**Data Availability Statement:** The data presented in this study are available on request from the corresponding author. The data are not publicly available due to confidentiality.

**Acknowledgments:** Our deepest gratitude goes to the editors and anonymous reviewers for their careful work and thoughtful suggestions that have helped to substantially improve this paper.

**Conflicts of Interest:** The authors declare no conflict of interest.

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
