# Peer review of "Effects of Coal Gangue on the Hydrochemical Components under Different Types of Site Karst Water in Closed Mines"

_water, doi:10.3390/w14193110_

Round 1
Reviewer 1 Report
Water
Manuscript Number: 1867182
Title: Effects of Coal Gangue on the Hydrochemical Components Under Different Types of Site Karst Water in Closed Mines
Type: Article
Keywords: Coal gangue ∙ Mine goaf ∙ Water quality parameters ∙ Cluster analysis ∙ Ions correlations
The main purpose of the manuscript Water-1867182 is to study the effect of coal gangue immersion on different types of karst water hydro-chemical composition using experimental methods of water rock immersion.
The paper appears no well-structured and some sections must be improved. Therefore, I believe the manuscript should be published only after major revision.
Comment
The manuscript does not contain a description of the chemical characteristics of the water, but only the chemical-physical parameters are reported. Furthermore, a geochemical classification of the waters studied by means of triangular diagrams coupled to a TIS is completely absent!
Specific comments
Line 97: change Water quality type in water chemical type
Line 106: change Water quality type in water chemical type
Line 135: the water sampling procedure is not reported
Line 170: How were the anions analyzed? The methodology you indicated is fine only for cations and metals, not for anions ...
Line 170: the alkalinity analysis is missing!
Line 177: insert a paragraph on the geochemical classification of analyzed water. I suggest to use two triangular diagrams and a TIS (salinity diagram), as proposed by:
Apollaro, C., Di Curzio, D., Fuoco, I., Buccianti, A., Dinelli, E., Vespasiano, G., Castrignanò, A., Rusi, S., Barca, D., Figoli, A. and Gabriele, B., 2022. A multivariate non-parametric approach for estimating probability of exceeding the local natural background level of arsenic in the aquifers of Calabria region (Southern Italy). Science of the Total Environment, 806, p.150345.
Line 271: The sulfate, could derive from pyrite, see the works of Fuoco et al. (2022) and Fang et al.(2022)
Fuoco I., De Rosa R., Barca D., Figoli A., Gabriele B. and Apollaro C., (2022). Arsenic polluted waters: Application of geochemical modelling as a tool to understand the release and fate of the pollutant in crystalline aquifers. Journal of Environmental Management, 301, p.113796.
Fan, R., Qian, G., Li, Y., Short, M.D., Schumann, R.C., Chen, M., Smart, R.S.C. and Gerson, A.R., 2022. Evolution of pyrite oxidation from a 10-year kinetic leach study: Implications for secondary mineralisation in acid mine drainage control. Chemical Geology, 588, p.120653.
Line 273: The fluorine in solution, come also from biotite dissolution as shown by Fuoco et al. (2022):
Fuoco, I., et al. "Use of reaction path modelling to investigate the evolution of water chemistry in shallow to deep crystalline aquifers with a special focus on fluoride." Science of The Total Environment 830 (2022): 154566.
Line 276: this is not true, because if we choose distance 5, these are not the only variables ...
Line 283: this is not true, because if we choose distance 10, these are not the only variables ...
Line 310: is this sentence another title of the paragraph?
Line 311: the TDS lettering is cut off
Line 335: was the mineral (Mirabilite) found in the area under investigation? if so, add a geological / mineralogical citation
Discussions and conclusions need to be rewritten taking into account previous comments
English must be reviewed by a native speaker
Recommended works must be added in the bibliography:
Fuoco, I., De Rosa, R., Barca, D., Figoli, A., Gabriele, B. and Apollaro, C., 2022. Arsenic polluted waters: Application of geochemical modelling as a tool to understand the release and fate of the pollutant in crystalline aquifers. Journal of Environmental Management, 301, p.113796.
Fuoco, I., et al. "Use of reaction path modelling to investigate the evolution of water chemistry in shallow to deep crystalline aquifers with a special focus on fluoride." Science of The Total Environment 830 (2022): 154566.
Apollaro, C., Di Curzio, D., Fuoco, I., Buccianti, A., Dinelli, E., Vespasiano, G., Castrignanò, A., Rusi, S., Barca, D., Figoli, A. and Gabriele, B., 2022. A multivariate non-parametric approach for estimating probability of exceeding the local natural background level of arsenic in the aquifers of Calabria region (Southern Italy). Science of the Total Environment, 806, p.150345.
Fan, R., Qian, G., Li, Y., Short, M.D., Schumann, R.C., Chen, M., Smart, R.S.C. and Gerson, A.R., 2022. Evolution of pyrite oxidation from a 10-year kinetic leach study: Implications for secondary mineralisation in acid mine drainage control. Chemical Geology, 588, p.120653.
Author Response
Dear Reviewer or Editor:
Thank you for the comments concerning our manuscript, No. water-1868914. Those comments are all valuable and very helpful for revising and improving our paper, as well as the important guiding significance to our research. We have studied comments carefully and have made correction which we hope meet with approval. Furthermore, according to Water Editor’s suggested, we to the main corrections in the paper and the responds to the reviewer’s comments are as follows:
Responds to Reviewer #1:
1.Response to comment: (Line 97: change Water quality type in water chemical type; Line 106: change Water quality type in water chemical type)
Response: Thank you very much for your comment, according to your comment, we have modified " Water quality type " to be " water chemical type ".
- Response to comment: (Line 135: the water sampling procedure is not reported)
Response: Thank you very much for your comment, according to your comment, we have modified the section, deleted the water sampling procedure.
- Response to comment: (Line 170: How were the anions analyzed? The methodology you indicated is fine only for cations and metals, not for anions ...)
Response: Thank you very much for your comment, according to your comment, we have modified the section.
- Response to comment: (Line 170: the alkalinity analysis is missing!)
Response: Thank you very much for your comment, because the test progress was frequent, according to the actual experimental situation, only part of the ions was tested, and many ions content (such as HCO3- ions) was not determined.
- Response to comment: (Line 177: insert a paragraph on the geochemical classification of analyzed water. I suggest to use two triangular diagrams and a TIS (salinity diagram), as proposed by:)
Response: Thank you very much for your comment. Your recommendations apply to studies with multiple samples, this article does not have multiple study samples, mainly analyzes the change of water quality indicators with soaking time. Your recommended method is very good, in future research we will use your suggestions for experimentation, thank you!
- Response to comment: (Line 271: The sulfate, could derive from pyrite, see the works of Fuoco et al. (2022) and Fang et al. (2022))
Response: Thank you very much for your comment. According to Fuoco et al. (2022) and Fang et al. (2022) we can know the sulfate could derive from pyrite. However, the clustering results showed that Pyrite and TDS, ED were grouped together in FGU immersion solution, reflecting a stronger correlation. Based on your comments, we have revised the article, thank you!
- Response to comment: (Line 273: The fluorine in solution, come also from biotite dissolution as shown by Fuoco et al. (2022))
Response: Thank you very much for your comment, because the experimental conditions are limited, all the influencing factors cannot be considered, and the experimental process did not consider the influence of the biotite dissolution on F- content.
- Response to comment: (Line 276: this is not true, because if we choose distance 5, these are not the only variables ...Line 283: this is not true, because if we choose distance 10, these are not the only variables ...)
Response: Thank you very much for your comment, we have modified these sections. According to the clustering conditions of R-type clusters, different immersion solutions can select different group distances as the criteria for selecting cluster objects.
- Response to comment: (Line 310: is this sentence another title of the paragraph?)
Response: Thank you very much for your comment, we have modified the section, it exists as a three-level heading.
- Response to comment: (Line 311: the TDS lettering is cut off)
Response: Thank you very much for your comment, we have modified the Figure.
- Response to comment: (Line 335: was the mineral (Mirabilite) found in the area under investigation? if so, add a geological / mineralogical citation)
Response: Thank you very much for your comment. The mineral composition contained in gangue is inferred from correlation and clustering results, and XRD results do not show the presence of Mirabilite, possibly due to the fact that XRD does not detect minerals with content below 3%.
Reviewer 2 Report
Comments on Manuscript (water-1867182)
Dear Authors
Generally, this paper discusses the effect of coal gangue on groundwater chemistry and quality in three mining sites. The authors simulate the natural hydrochemical processes in the laboratory using a static immersion test. The article is interesting, and it reflects an actual case study. However, the article needs enhancement to be readable.
I reviewed the manuscript line by line, and for better enhancement of the Manuscript, please find my comments as follows:
1. Abstract: authors should enhance the abstract by adding the aims, approach, and main findings.
2. Introduction: It is recommended that the authors enhance the Ordovician limestone aquifer section with references and lithology illustrations.
a. Figure 1: caption above figure. It should be below.
b. No caption for the stratigraphic column.
3. Material and methods: It is recommended that authors summarize the sampling and preparation section to be readable.
4. Results and Discussions: This large section makes it difficult for readers to follow up. So, I recommend authors to add a summary of the main results at the end of this section.
The subsection: EC correlation analysis of each immersion solution. I think the result in this subsection is similar to the TDS section. So, it is recommended to exclude it.
In addition, it would be good if the author could add the saturation indices of the groundwater samples. This will help to support their interpretation in terms of mineral dissolution.
If possible, I recommend that the authors run an XRD analysis of the solid samples to obtain the mineral composition.
5. Conclusion: the authors should emphasize the main hydrochemical processes and dominant minerals that control water chemistry. It is recommended to highlight the difference in the effect of coal gangue on groundwater quality between the three mine sites.
Author Response
Dear Reviewer or Editor:
Thank you for the comments concerning our manuscript, No. water-1868914. Those comments are all valuable and very helpful for revising and improving our paper, as well as the important guiding significance to our research. We have studied comments carefully and have made correction which we hope meet with approval. Furthermore, according to Water Editor’s suggested, we to the main corrections in the paper and the responds to the reviewer’s comments are as follows:
Responds to Reviewer #2:
- Response to comment: (1. Abstract: authors should enhance the abstract by adding the aims, approach, and main findings.)
Response: Thank you very much for your comment, according to your comment, we have modified the summary by adding goal, method and key findings.
- Response to comment: (2. Introduction: It is recommended that the authors enhance the Ordovician limestone aquifer section with references and lithology illustrations. a. Figure 1: caption above figure. It should be below. b. No caption for the stratigraphic column.)
Response: Thank you very much for your comment, according to your comment, we have modified the position of the caption. In addition, we have added relevant references about the Ordovician limestone aquifer section.
- Response to comment: (3. Material and methods: It is recommended that authors summarize the sampling and preparation section to be readable.)
Response: Thank you very much for your comment, according to your comment, we have summarized the sampling and preparation section.
- Response to comment: (4. Results and Discussions: This large section makes it difficult for readers to follow up. So, I recommend authors to add a summary of the main results at the end of this section.
The subsection: EC correlation analysis of each immersion solution. I think the result in this subsection is similar to the TDS section. So, it is recommended to exclude it.
In addition, it would be good if the author could add the saturation indices of the groundwater samples. This will help to support their interpretation in terms of mineral dissolution.
If possible, I recommend that the authors run an XRD analysis of the solid samples to obtain the mineral composition.)
Response: Thank you very much for your comment, according to your recommend, we added a summary of the main results at the end of every section. According to your recommend, we deleted the EC section. In addition, we supplement XRD analysis of the solid samples and obtain the mineral composition. Because the experiment process did not observe HCO3- data, the saturation indices of the groundwater samples calculation cannot be added.
- Response to comment: (5. Conclusion: the authors should emphasize the main hydrochemical processes and dominant minerals that control water chemistry. It is recommended to highlight the difference in the effect of coal gangue on groundwater quality between the three mine sites.)
Response: Thank you very much for your comment, according to your comment, we have modified the conclusion.
We tried our best to improve the manuscript and made some changes in the manuscript. These changes will not influence the content and framework of the paper. And here we did not list the changes throughout the paper.
We appreciate for Editor/Reviewers’ warm work earnestly, and hope that the correction will meet with approval.
Once again, thank you very much for your comments and suggestions.
Yours sincerely,
Ji, M. Sc.
Round 2
Reviewer 1 Report
regarding the origin of the fluoride, the authors reply that:
Thank you very much for your comment, because the experimental conditions are limited, all the influencing factors cannot be considered, and the experimental process did not consider the influence of the biotite dissolution on F- content
Despite this, in the literature it is known that fluoride can also derive from micas (see the work of Fuoco et al 2022 and cited bibliography) and therefore it must be taken into account !!!
Reference:
Add:
Fuoco, I., et al. "Use of reaction path modelling to investigate the evolution of water chemistry in shallow to deep crystalline aquifers with a special focus on fluoride." Science of The Total Environment 830 (2022): 154566.
Author Response
Dear Reviewer:
Thank you for the comments concerning our manuscript, No. water-1868914. According to your suggestions, we have added relevant references to the article.
We appreciate for your warm work earnestly, and hope that the correction will meet with approval.
Once again, thank you very much for your comments and suggestions.
Yours sincerely,
Xu, Prof.
Reviewer 2 Report
The manuscript has been improved and the authors incorporated all comments in the revised manuscript.
Author Response
Thank you for the comments concerning our manuscript!